# Effect of Knotweed in Diet on Physiological Changes in Pig



Petr Maděra [1],*, Marcela Kovářová [1], Tomáš Frantík [1,2], Radek Filipčík [3], Jan Novák [4], Štěpán Vencl [5], Lucie Maděrová [6], Miroslav Rozkot [7], Stanislava Kuchařová [7], Eva Václavková [7], Jana Truněčková [7], Jana Volková [8], Zora Nývltová [8] and Michal Bartoš [8]

[1] Department of Forest Botany, Dendrology and Geobiocoenology, Mendel University in Brno, Zemědělská 1665/3, 613 00 Brno, Czech Republic; mkvr@seznam.cz (M.K.); tomas.frantik@seznam.cz (T.F.)
[2] Institute of Botany of the CAS, 252 43 Průhonice, Czech Republic
[3] Department of Animal Breeding, Mendel University in Brno, Zemědělská 1665/1, 613 00 Brno, Czech Republic; radek.filipcik@mendelu.cz
[4] Dibaq, a.s., Helvíkovice 90, 564 01 Žamberk, Czech Republic; novakjan@dibaq.cz
[5] Jirchářská 217, 517 41 Kostelec nad Orlicí, Czech Republic; mobilnivet@gmail.com
[6] Faculty of Veterinary Hygiene and Ecology, University of Veterinary and Pharmaceutical Sciences Brno, Palackého tř. 1946/1, 612 42 Brno, Czech Republic; lucymaderova@seznam.cz
[7] Institute of Animal Sciences p.r.i., Department of Pig Breeding, Komenského 1239, 517 41 Kostelec nad Orlicí, Czech Republic; rozkot.miroslav@vuzv.cz (M.R.); kucharova.stanislava@vuzv.cz (S.K.); vaclavkova.eva@vuzv.cz (E.V.); truneckova.jana@vuzv.cz (J.T.)
[8] Research Institute of Organic Synthesis, Rybitví 296, 533 54 Rybitví, Czech Republic; jana.volkova@vuos.com (J.V.); zora.nyvltova@vuos.com (Z.N.); michal.bartos@vuos.com (M.B.)
* Correspondence: petrmad@mendelu.cz; Tel.: +420-739-341-962

**Abstract:** Knotweeds (*Reynoutria* spp.) are plants producing useful secondary metabolites, including stilbenes (resveratrol and piceid have been studied more thoroughly) and emodin. Many studies have shown the positive effects of resveratrol on the health status of humans and animals. Resveratrol has been added into pigs' diet as a pure extract, but it has never been supplemented into the fodder with knotweed biomass which contains other secondary metabolites, thus we would expect it would provide a more complex effect. The study objective is to discover whether the 2 weight percent addition of knotweed into pigs' diet will have positive effects on their health. We compared two groups of Prestice Black-Pied pigs, the experimental group was fed by fodder with the knotweed rhizomes additive, the control group without knotweed additive. Investigated parameters were feed consumption, the composition of excrements, weight increment, muscle-to-fat ratio, fatty acid composition and blood haematology and biochemistry. The addition of knotweed stimulated a whole range of physiological changes. It positively stimulated weight growth and increased the back fat and proportion of muscle, but statistically significant only in gilts. On the other hand, the changes in fatty acid composition seemed to be unsatisfactory. It is the first study of the effects of knotweed on pigs' development, and more detailed research is desirable.

**Keywords:** Reynoutria; fodder additive; health condition; Prestice Black-Pied pig

## 1. Introduction

Appropriate and balanced fodder is crucial in the pig diet. Studies focusing on using plants or plant parts as fodder additives for pigs are not frequently executed. In one instance, Kafantaris et al. [1] tested a grape filling as a fodder additive for weaned piglets in order to examine the effect on pigs' well-being, productivity and meat quality. Another experiment with fodder was performed by Adebiyi et al. [2], who fed weaned piglets hydroponic corn fodder. The results showed that the introduction of hydroponic corn fodder into the pig diet improved the nutrient digestibility of weaned pigs. Václavková and Bělková [3] described an experiment with Prestice Black-Pied pigs that examined the effect of linseed meal in a pigs' feed ration on meat quality. So far, we have not found any

indication in the literature that knotweed (*Reynoutria* spp.) has been tested as a fodder additive in a pig diet.

The Prestice Black-Pied pig is a Czech breed that is maintained under the National Program for the Conservation and Utilisation of Genetic Resources. The Prestice Black-Pied breed originates in western Bohemia in the region of the towns Přeštice, Domažlice and Klatovy. It is characterised by good reproductive performance and adaptability for breeding conditions and nutrition. Under normal intensive fattening conditions, it is characterised by a less favorable carcass value and cannot be compared with the performance of modern meat breeds and hybrids [4]. Prestice Black-Pied pigs show higher content of intramuscular fat and backfat thickness, higher pH value, lower lean meat content and drip loss value than commercial hybrid pigs [5]. Compared to improved breeds, the Prestice pig is characterised by lower growth performance and higher carcass fatness; and is therefore not competitive under large-scale rearing conditions. On the other hand, so-called "primitive" characteristics, such as hardiness and adaptability, have been preserved in this breed. Thus, they are suitable for extensive rearing conditions similar to other indigenous European breeds [6].

Japanese knotweed (*Reynoutria japonica*), Sakhalin knotweed (*R. sachalinensis*) and their hybrid Bohemian knotweed (*R. × bohemica*) are considered some of the most dangerous invasive plant species in the temperate zone of the Northern Hemisphere [7–9]. However, knotweeds are important and useful plants: they are a source of biomass energy [10,11], and they are also grown for ornamental purposes [12,13] or as fodder [14].

These plants produce useful secondary metabolites, including stilbenes (of which resveratrol and piceid have been studied more thoroughly) and emodin [15–17]. Resveratrol is a plant polyphenol [18], piceid is a resveratrol glycoside [19,20] and emodin is a natural anthraquinone derivate [21].

In vitro tests with the above-stated substances on laboratory animals and clinical studies in human medicine have shown antibacterial effects [22,23], antifungal effects [24–28], antiparasitic effects [29], anti-inflammatory effects [30], anticancer agents [31–42] and effects against vascular and other diseases of civilisation including those related to obesity and diabetes [43–46]. Antiaging effects were observed with resveratrol supplementation [18], and piceid was found to protect heart cells damaged by lack of oxygen and glucose and also to prevent platelet aggregation after the administration of clonidine, a medicine to reduce blood pressure [19,20]. Emodin may also ameliorate oxidative stress [47].

Resveratrol has been added to pigs' diet, leading to a reduction of body fat [44,48,49]. These studies proved that a diet supplemented by resveratrol may reduce the depth of pigs' back fat in the last phase of fattening, resulting in an overall reduction of subcutaneous fat [50]. Results suggest that resveratrol is an effective supplement for increasing the quality of pork [51]. Cheng et al. [52] also proved that resveratrol added in different concentrations into pigs' diet may improve serum lipid profiles and reduce body fat storage. The effects of resveratrol were also monitored in rodents. It was found that resveratrol reduced the body fat of rodents that were fed a high-calorie diet [48,53,54].

As is obvious from the above-stated summary, resveratrol and other secondary metabolites contained in knotweed biomass have a positive influence on human and animal health. The objective of the submitted study was to discover whether the addition of knotweed into pigs' diet will have the same positive effects as the addition of pure extract of resveratrol. We especially expected an effect on muscle to fat ratio and on lipid metabolism.

## 2. Materials and Methods

### 2.1. Experimental Design

The experiment was performed at the Institute of Animal Science in Kostelec nad Orlici in 2019. The Prestice Black-Pied breed was used for the experiment.

Piglets from four sows of the Prestice breed impregnated by three fathers (boars SC 165, PIT293 and WSN56) were used in the experiment. Sow No. 2612 bore 10 piglets on

19 May 2019; sow No. 2343 bore 9 piglets on 28 May 2019; sow No. 2128 bore 5 piglets on 25 May 2019 and; sow No. 2553 bore 7 piglets on 25 May 2019.

Twenty-four hours after birth, the piglets were weighed and preliminarily marked by a serial number for identification. At the age of seven days, they were marked by coloured labels with the breeding number and were injected with Fe. Each litter was divided into two groups according to birth weight and sex; thus, eight groups of piglets, four control (16 piglets in total, 8 boars and 8 gilts) and four experimental (15 piglets in total, 8 boars and 7 gilts) were created. All boars were castrated. On 20 June 2019, all piglets were weaned, and blood samples were taken from them. The day after blood was taken, they were given Shotapen antibiotics, and three days following this, they were given Suvaxyn for active immunisation in order to reduce the occurrence of lung lesions caused by Mycoplasma hyopneumoniae infection. Throughout the experiment, the animals were not manipulated; they stayed in the same stalls all the time. Therefore, they did not come into contact with other pigs, and they were not disturbed in any way. Throughout the experiment, food intake was steady. The pigs were housed in stalls with grates over a solid floor. After weaning, each group was fed separately in their own stalls, and their health condition was monitored. At the end of the experiment, a parasitological examination was performed in each stall. All examinations were negative. All piglets gained weight continuously and did not suffer from diarrhea at any time.

### 2.2. Fodders and Feeding Regime

The below-stated fodders (FC) were produced by Dibaq (The Czech Republic, traditional producer of feed for domestic and farm animals, www.dibaq.cz) and used in the experiment:

- Selvico (prestarter) is a complete feed designed for piglets up to 8 kg. It is served 7–14 days before weaning to secure the habit and then 10–14 days after weaning. It contains barley, wheat, genetically modified (GM) soybean meal, dried rhizome of Bohemian knotweed, lactose, secondary products from sweets manufacturing, soy oil, sugar beet pulp, fish meal, soy protein, glucose and dried poultry blood.
- ČOS = EW (Early Weaning) is a complete feed designed for the nutrition of piglets up to 18 kg. It contains barley, wheat, soybean meal GM, dried poultry blood, bran, fish meal, dried rhizome of Bohemian knotweed, lactose, fish oil, calcium dihydrogen phosphate and calcium carbonate.
- A1 (subsequent mixture) is a complete feed intended for the fattening of pigs up to 35 kg. It contains barley, wheat, soybean meal GM, bran, sugar beet pulp, dried rhizome of Bohemian knotweed, soy oil, calcium carbonate, sodium chloride, calcium dihydrogen phosphate and dried poultry blood.
- KPB is a commonly used mixture for pregnant sows, but in our case, we used it to feed pigs from 35 kg to the carcass weight. It contains barley, wheat, soybean meal GM, dried poultry blood, bran, fish meal, dried rhizome of Bohemian knotweed, fish oil, lactose, calcium dihydrogen phosphate and calcium carbonate.

Fodders with the additive of knotweed were enriched with 2 weight percent of dry rhizomes of Bohemian knotweed. Thus, they contained 110.2 mg of piceid, 8.1 mg of oxyresveratrol, 35.6 mg of resveratrol, 1.4 mg of pterostilbene and 77.9 mg of emodin per kilo of fodder. So, the average daily dose of resveratrol was 226.82 mg per animal during the duration of the experiment if we take into account also the resveratrol content in other substances, i.e., piceid (glycoside of resveratrol) and oxyresveratrol. However, the daily doses were changed according to the feed consumption during the pigs' development.

The experiment began after the piglets were weaned. The piglets were fed Selvico FC from their tenth day of age. They were then fed EW FC for 30 days, followed by A1 FC for 28 days, followed by KPB FC until the end of the experiment. Piglets from control groups were given feed compounds with standard ingredients; the experimental groups received the same feed compounds with the addition of knotweed.

*2.3. Measured Characteristics*

2.3.1. Feed Consumption

The fodder was always served once a day at the same time throughout the whole experiment ad libitum. Feed consumption was recorded every day for each experimental and control group.

2.3.2. Composition of Excrements

Excrements were taken continuously. A mixed sample of excrements was created for each experimental and control group and fodder type and analysed together with a fodder sample. The dry matter contents of the excrements were specified, and analyses of the content of ash, fats, fibre, nitrogen and nitrogenous substances were performed.

2.3.3. Weight of Piglets

The piglets were weighed once a month throughout the experiment until 19 November 2019.

2.3.4. Muscle-To-Fat Ratio

At the end of the experiment, the muscle-to-fat ratio was measured using a Mindray (Shenzhen, China) ultrasonic device.

After the experiment, the animals were slaughtered at an experimental slaughterhouse of the IAS in Uhrineves, Prague. For capacity reasons, this was done in two groups on 3 December and 10 December 2019. A measurement of the muscle-to-fat ratio in each carcass was then performed via the ZP method [55], the backfat height was evaluated, values of pH1 and pH24 were measured and, 24 h post mortem, samples of M. longissimus dorsi and backfat were taken. The samples were analysed to determine the share of intramuscular fat and the fatty acid composition of the meat and lard. The intramuscular fat content was determined according to CSN ISO 1444 [56] by extraction in the Soxtec 1043 apparatus (FOSS Tecator AB, Hoganas, Sweden). Muscle pH was measured using a portable pH meter (Testo 205) equipped with a glass electrode at 45 min and 24 h post-mortem in fresh samples. The fatty acid composition of the meat was determined after chloroform–methanol extraction of total lipids [57]. Fatty acid methyl esters were prepared in accordance with CSN ISO 5509 [58] and analysed by gas chromatography (gas chromatograph 6890N Agilent Technologies) according to CSN ISO 5508 [59].

2.3.5. Haematology and Biochemistry

Blood sampling was done at the place of breeding at the beginning and the end of the experiment as a part of standard care. The samples were taken from the vena auricularis by a veterinary doctor. One mL of blood was taken in a test tube with EDTA3K for haematological examination, and 10 mL of blood was taken in a test tube with a procoagulant gel. The samples were then immediately stored at refrigerator temperature until their elaboration.

After delivery, the biochemical samples were immediately submitted to a laboratory that performed a biochemical examination for the following parameters: total serum protein, urea, serum creatinine, serum cholesterol, serum glucose, AST, ALT, ALP, GMT, direct serum bilirubin and total serum bilirubin.

The haematologic samples were elaborated via a veterinary haematology analyser for a blood count determination; at the same time, blood smears with their subsequent fixation and haematological stains using Hemacolour dye were performed. The calculations of individual cellular elements, i.e., neutrophils, eosinophils, basophils, monocytes, lymphocytes and active lymphocytes, were performed using the LeukoCounter mobile application. The total number of cells was 200, and individual populations of white blood cells were determined as a percentage.

### 2.3.6. Statistical Data Evaluation

Two-way ANOVA (factors: Knotweed and Sex or Knotweed and Feed) with interaction was applied to the data. For the comparison of the experimental and control treatment, ANOVA with a nested factor (litter) was used. If homoscedasticity or normality of the data was not met, the non-parametric Mann-Whitney test was applied. All statistical analyses were performed using STATISTICA v.12 software. The variance of the data was represented by the standard error of mean. The level of significance $p = 0.05$ was used whenever not mentioned.

## 3. Results

### 3.1. Fodder Consumption and Digestibility

Overall, no significant difference in fodder consumption was found with or without knotweed (Table 1), with no significant differences in fodder consumption per 1 kg increment between the experimental and control groups.

**Table 1.** The fodder consumption of the experimental group of pigs (WK) and the control group of pigs (NK).

| Experimental Group | ČOS (EW) [kg] | A1 [kg] | KPB [kg] |
|---|---|---|---|
| NK (*n* = 16) | 715.8 | 826.5 | 3287.0 |
| WK (*n* = 15) | 810.0 | 900.0 | 3477.5 |
| Feeding duration | 2019.6.20–2019.8.12 | 2019.8.13–2019.9.9 | 2019.9.10–2019.12.8 |

Knotweed added to fodder affected the composition of excrements (Figure 1). A statistically significant reduction of fat in faeces was recorded in animals feeding on EW (ČOS). When feeding on A1, the fibre and dry matter content in the faeces were significantly reduced. When feeding on KPB, the fibre in the faeces was significantly reduced and, on the other hand, nitrogenous substances and ash contents in the faeces significantly increased.

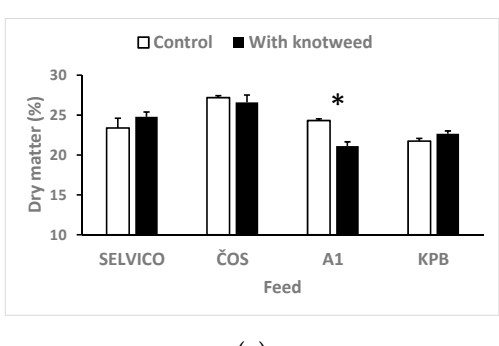

(a)

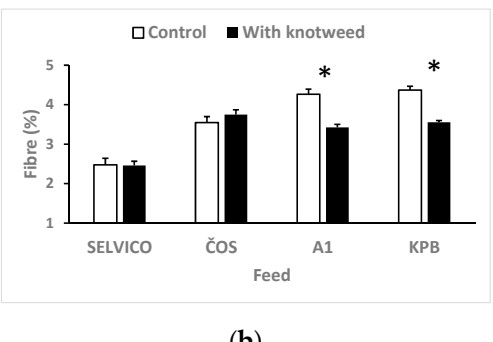

(b)

**Figure 1.** *Cont.*

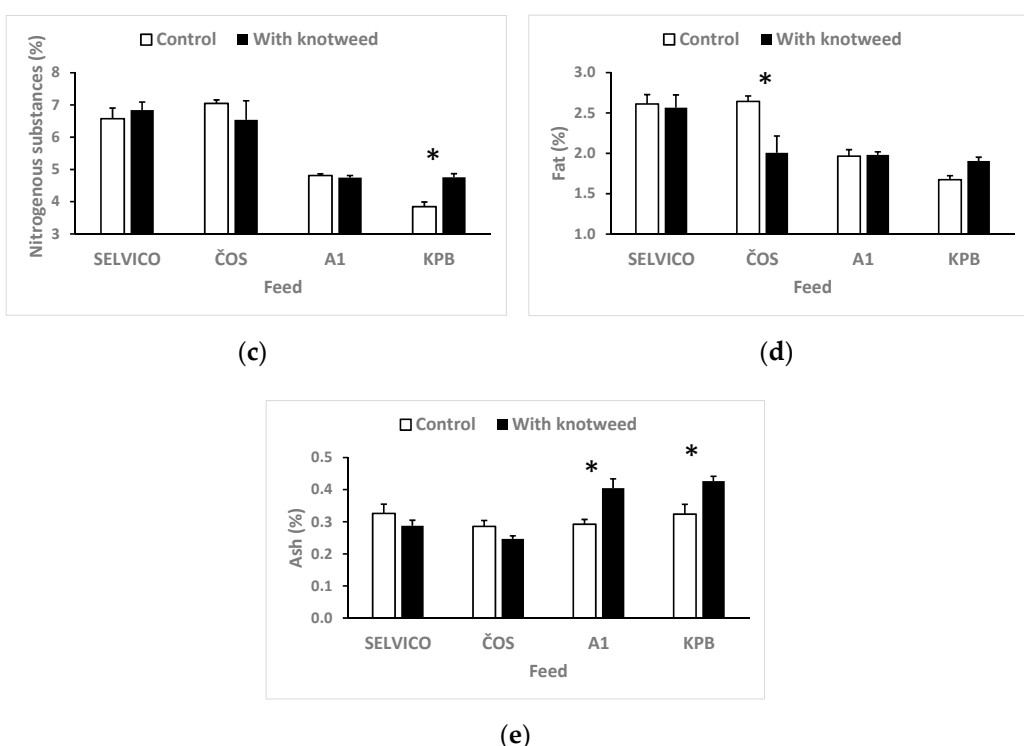

(c)

(d)

(e)

**Figure 1.** Comparison of the composition of faeces (**a**—dry matter, **b**—fibre, **c**—nitrogenous substances, **d**—fat, **e**—ash) for individual fodder types between animals fed on fodder with knotweed and control groups fed on standard fodder. Means + SEM. Difference significant on $p < 0.05$ marked by an asterisk.

### 3.2. Total Weight and Increments

The introduction of Bohemian knotweed into a feed mixture for pigs influenced the growth ability of gilts and boars. Gilts fed on a feed mixture with knotweed showed significantly ($p < 0.05$) higher growth intensity, while for boars, the opposite trend was recorded ($p > 0.05$) (Figure 2), thereby decreasing the weight difference between sexes, which was significant at the beginning of the experiment.

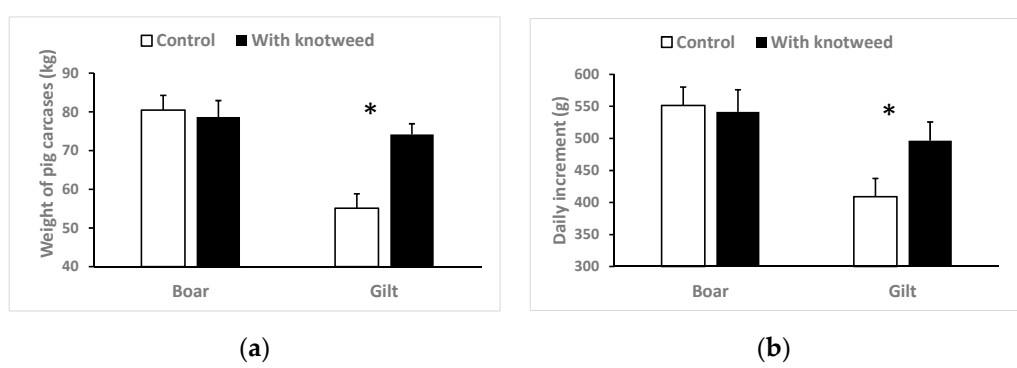

(a)

(b)

**Figure 2.** Weight of pig carcasses (boars and gilts) at the end of the experiment (**a**) and the average daily weight increment since birth to 19 November 2019 (**b**). Means + SEM. Difference significant on $p < 0.05$ marked by an asterisk.

### 3.3. The Muscle-To-Fat Ratio and Fatty Acid Content

The ultrasonic measurement showed the negative effect of knotweed on the percentual content of lean meat; however, this was strongly influenced by the sex of the pigs. While there was no significant difference in the lean meant content between the experimental and control groups in boars ($p > 0.05$), in sows, the lean meat content was 1.5 to 2.8% higher for gilts from the control group (Figure 3).

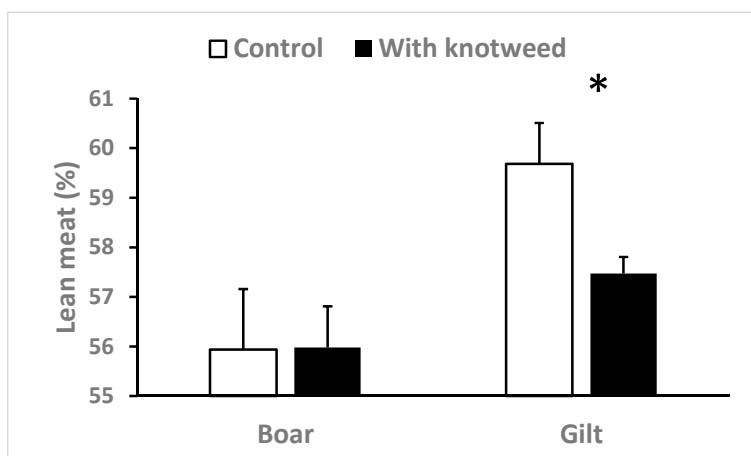

**Figure 3.** The content of lean meat (muscle) measured by ultrasound in pigs fed on a feeding mixture with the knotweed additive in comparison to pigs fed on a common feeding mixture, separated according to sex. Means + SEM. Difference significant on $p < 0.05$ marked by an asterisk.

In accordance with the higher recorded weight of gilts fed on knotweed (Figure 2), a greater amount of back fat and muscle (Figure 4) was recorded for the same group compared to a control group using a thorough analysis of a carcass. Knotweed feeding positively affected the amount of back fat in gilts; the average height of lard was 15.4 mm, whilst in boars, the back fat height was around 22.75 mm. No significant ($p > 0.05$) difference in the height of back fat between boars from the control and experimental group was recorded; however, in gilts from the control group, back fat was about 5.15 mm lower ($p < 0.05$) than in gilts from the experimental group (Figure 4). Thus, the reduced content of lean meat in gilts fed on knotweed was caused by the higher quantity of lard (50.24%) rather than by the quantity of muscle (13.10%).

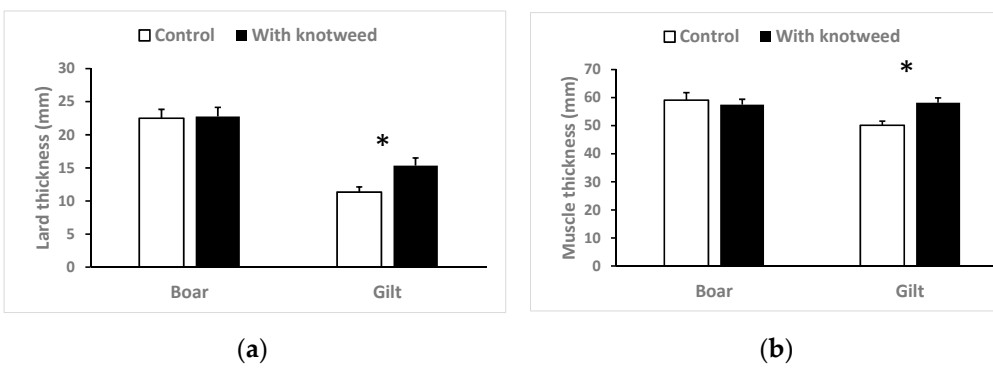

(**a**)                                       (**b**)

**Figure 4.** The average thickness of lard (**a**) and muscle (**b**) measured at slaughter using the ZP method in pigs fed on a feeding mixture with knotweed addition compared to pigs fed on a standard feeding mixture, separated according to sex. Means + SEM. Difference significant on $p < 0.05$ marked by an asterisk.

The introduction of knotweed into feeding mixtures affected the content of intramuscular fat. Meat from the experimental group contained $1.96 \pm 0.06\%$ intramuscular fat on average, while in the control group, a significantly ($p < 0.05$) higher content of intramuscular fat ($2.11 \pm 0.05\%$) was recorded.

Individual fatty acid contents were specified in extracted fat (C6:0–C24:0). In terms of the usability of individual fatty acids by humans, not only is the fatty acid content important but effective usage also depends on the ratios between each acid or groups of acids. For this reason, fatty acids (FA) were divided into saturated fatty acids (SFA), monounsaturated fatty acids (MUFA) and polyunsaturated fatty acids (PUFA), and their ratios were calculated (Table 2). The presence of SFA was found in the intramuscular

fat ($42.63 \pm 0.3$ g $100$ g$^{-1}$), and an inconclusive higher ratio of SFA was found in gilts ($42.37 \pm 0.86$ g $100$ g$^{-1}$) and boars ($44.07 \pm 0.23$ g $100$ g$^{-1}$) from the control group (Table 1). This was caused mainly by a significantly ($p < 0.05$) higher ratio of caproic acid (C6:0), lauric acid (C12:0) and stearic acid (C18:0). The highest ratio ($46.06 \pm 0.40$ g $100$ g$^{-1}$) of monounsaturated fatty acids was found in the intramuscular fat, and no significant difference ($p > 0.05$) between gilts and boars in the control and experimental groups was discovered. Significant ($p < 0.01$) differences were found in the content of polyunsaturated fatty acids, which are the most important FAs for humans. It is possible to state that the implementation of knotweed into a feeding mixture for pigs caused a reduction of PUFA content in favour of MUFA; highly significant ($p < 0.01$) differences were found between the control ($10.91 \pm 0.56$ g $100$ g$^{-1}$) and the experimental group ($14.91 \pm 1.94$) of gilts. In boars, the trend was similar although statistically insignificant ($p > 0.05$) (Table 3).

**Table 2.** The composition of fatty acids in the intramuscular fat of *musculus longissimus lumborum et thoracic* (MLLT) and in the depot fat of pigs. The significant difference between the experimental group of pigs (WK) and the control group of pigs (NK) marked by an asterisk. (Means $\pm$ SEM g $100$ g$^{-1}$).

|  | MLLT | | Backfat | |
|---|---|---|---|---|
|  | WK | NK | WK | NK |
| SFA | $43.34 \pm 0.44$ * | $41.90 \pm 0.46$ | $42.61 \pm 0.50$ | $41.28 \pm 0.43$ |
| nonSFA | $56.66 \pm 0.44$ * | $58.10 \pm 0.46$ | $57.39 \pm 0.50$ | $58.72 \pm 0.43$ |
| MUFA | $46.88 \pm 0.45$ * | $45.25 \pm 0.61$ | $44.38 \pm 0.27$ * | $41.62 \pm 0.60$ |
| PUFA | $9.78 \pm 0.43$ * | $12.85 \pm 0.61$ | $13.02 \pm 0.42$ * | $17.10 \pm 0.64$ |
| PUFA n-3 | $1.09 \pm 0.08$ * | $1.69 \pm 0.10$ | $1.37 \pm 0.05$ * | $2.28 \pm 0.12$ |
| PUFA n-6 | $8.69 \pm 0.37$ * | $11.16 \pm 0.51$ | $11.65 \pm 0.37$ * | $14.83 \pm 0.53$ |
| n-6/n-3 | $8.12 \pm 0.29$ * | $6.69 \pm 0.14$ | $8.56 \pm 0.18$ * | $6.68 \pm 0.30$ |

**Table 3.** The composition of fatty acids in the intramuscular fat MLLT and in the depot fat of pigs. The experimental group of pigs (WK) and the control group of pigs (NK) separated according to sex. (Means $\pm$ SEM g $100$ g$^{-1}$).

|  | MLLT | | | | Backfat | | | |
|---|---|---|---|---|---|---|---|---|
|  | Gilts | | Boars | | Gilts | | Boars | |
|  | WK | NK | WK | NK | WK | NK | WK | NK |
| SFA | $42.37 \pm 0.86$ | $41.25 \pm 0.49$ | $44.07 \pm 0.23$ | $42.41 \pm 0.70$ | $41.70 \pm 0.97$ | $41.41 \pm 0.52$ | $43.29 \pm 0.38$ | $41.18 \pm 0.68$ |
| MUFA | $46.71$ $^{Aa} \pm 0.99$ | $43.85$ $^{AB} \pm 0.64$ | $47.00$ $^A \pm 0.38$ | $46.30$ $^{Aa} \pm 0.79$ | $44.35$ $^{AB} \pm 0.26$ | $39.43$ $^B \pm 0.66$ | $44.40$ $^{AB} \pm 0.44$ | $43.26$ $^{Bb} \pm 0.25$ |
| PUFA | $10.91$ $^{AB} \pm 0.56$ | $14.91$ $^{CD} \pm 0.79$ | $8.93$ $^A \pm 0.45$ | $11.31$ $^{ACD} \pm 0.29$ | $13.95$ $^{BD} \pm 0.80$ | $19.16$ $^E \pm 0.50$ | $12.31$ $^{BC} \pm 0.25$ | $15.56$ $^D \pm 0.63$ |
| MUFA/PUFA | $4.35$ $^{Aba} \pm 0.28$ | $2.99$ $^{CEb} \pm 0.18$ | $5.36$ $^A \pm 0.28$ | $4.12$ $^{BDa} \pm 0.15$ | $3.23$ $^{CDb} \pm 0.17$ | $2.07$ $^E \pm 0.09$ | $3.62$ $^{BC} \pm 0.10$ | $2.81$ $^{CE} \pm 0.12$ |
| PUFA/SFA | $0.26$ $^{Aba} \pm 0.02$ | $0.36$ $^{CEb} \pm 0.02$ | $0.20$ $^A \pm 0.01$ | $0.27$ $^{Aa} \pm 0.01$ | $0.34$ $^{BC} \pm 0.03$ | $0.46$ $^{Ec} \pm 0.02$ | $0.28$ $^{AC} \pm 0.01$ | $0.38$ $^{CE} \pm 0.02$ |
| MUFA/SFA | $1.11 \pm 0.04$ | $1.06 \pm 0.02$ | $1.07 \pm 0.01$ | $1.10 \pm 0.04$ | $1.07 \pm 0.03$ | $0.95 \pm 0.02$ | $1.03 \pm 0.02$ | $1.05 \pm 0.02$ |
| PUFA n-3 | $1.30$ $^{Aa} \pm 0.13$ | $2.01$ $^{BCb} \pm 0.14$ | $0.94$ $^A \pm 0.04$ | $1.45$ $^{ABc} \pm 0.07$ | $1.43$ $^{AB} \pm 0.11$ | $2.60$ $^C \pm 0.09$ | $1.32$ $^A \pm 0.05$ | $2.04$ $^{Cd} \pm 0.15$ |
| PUFA n-6 | $9.61$ $^{AB} \pm 0.48$ | $12.90$ $^{CD} \pm 0.66$ | $8.00$ $^A \pm 0.41$ | $9.86$ $^{Abd} \pm 0.24$ | $12.53$ $^{CDc} \pm 0.70$ | $16.56$ $^E \pm 0.42$ | $10.99$ $^{BCa} \pm 0.22$ | $13.52$ $^{Db} \pm 0.49$ |
| n-6/n-3 | $7.60 \pm 0.57$ | $6.47$ $^B \pm 0.21$ | $8.51$ $^a \pm 0.23$ | $6.86$ $^b \pm 0.19$ | $8.84$ $^{Aa} \pm 0.20$ | $6.38$ $^{Bb} \pm 0.09$ | $8.35$ $^a \pm 0.26$ | $6.90 \pm 0.52$ |

a, b, c, d = $p < 0.05$; A, B, C, D, E = $p < 0.01$.

In the intermuscular fat, a statistically significant reduction in the ratios of n-6 PUFA and n-3 PUFA was recorded in the group of pigs fed on knotweed compared to the control group, and the n6/n3 PUFA ratio was significantly higher in the experimental group of pigs compared to the control group (Table 1). A significantly higher ratio of PUFA n-3 was recorded in gilts in the control group ($2.01 \pm 0.14$ g $100$ g$^{-1}$). The trend was similar in boars but without significant ($p > 0.05$) differences (Table 3).

A fatty acid content analysis was also performed on the back fat. The SFA ratio was in the range of variation from $41.18$ to $43.29$ g $100$ g$^{-1}$, and the measured values were comparable with the SFA ratio in the intermuscular fat. No significant differences ($p > 0.05$) were found between the control and experimental groups of boars and gilts in terms of MUFA representation in lard; however, it is possible to state that MUFA were eliminated in favour of PUFA thanks to the knotweed additive (Table 2). A significant ($p < 0.01$) difference was also found in the MUFA and PUFA ratio between back fat and

intramuscular fat. An interesting finding is the higher PUFA ratio in lard compared to intramuscular fat. The significantly ($p < 0.01$) highest PUFA contents were found in the lard of gilts ($19.16 \pm 0.50$ g 100 g$^{-1}$) and boars ($15.56 \pm 0.63$ g 100 g$^{-1}$) from the control group. Thanks to this fact, the MUFA/PUFA ratio was more favourable for the back fat. The same conclusions can be stated in the case of the PUFA/SFA ratio and the MUFA/SFA ratio (Table 3).

The content of n6 and n3 polyunsaturated fatty acids in the lard of pigs fed fodder with knotweed (Table 2) was statistically significantly lower than in the control group of pigs, and the n6/n3 ratio of polyunsaturated fatty acids in the lard of pigs fed fodder with knotweed (Table 2) was clearly statistically higher than in the control groups of pigs. In the case of the n-6/n-3 FA ratio, more favourable values for humans were found in the back fat of control groups of gilts ($6.38 \pm 0.09$) and boars ($6.90 \pm 0.52$) (Table 3).

### 3.4. Biochemistry and Haematology

From the whole range of measured biochemical and haematological parameters, only the ones with statistically significant differences between the experimental and control groups will be presented.

### 3.4.1. Total Serum Protein

A significant change in total serum protein (TSP) between the experimental animals and the control group was found at the end of the experiment. No significant difference between groups was recorded upon the first sampling. The same changes, i.e., an increase of the concentration, were observed in both groups, but the group fed on fodder with knotweed had a higher concentration of total protein. From other results, it is certain that this difference is due to an increase in the concentration of serum albumin, not globulins (Figure 5).

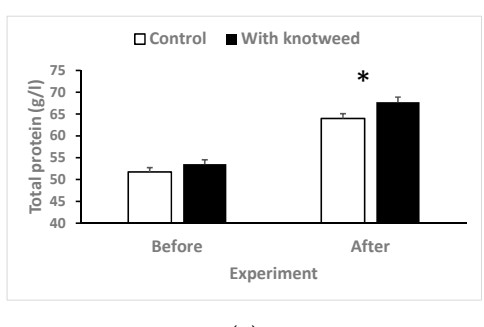
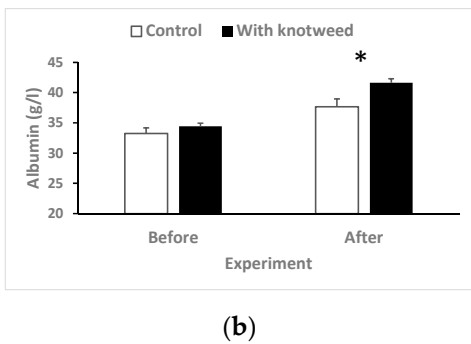

(**a**) (**b**)

**Figure 5.** Total serum protein (**a**) and serum albumin (**b**) in the experimental group of pigs fed on fodder with knotweed and the control group at the beginning and the end of the experiment. Means + SEM. Difference significant on $p < 0.05$ marked by an asterisk.

A significant change in total serum protein appeared in gilts. The results between gilts and boars (Figure 6) are very interesting because a significant statistical difference ($p = 0.02742$) was noted in the comparison of boars without knotweed and gilts without knotweed. This means we have proven that the change in concentration of TSP is related to the sex. If we compare boars fed on knotweed with gilts from the control group ($p = 0.00538$), we find an even greater statistical difference. However, when we compare boars that were not fed on knotweed to knotweed fed gilts, the statistical difference is insignificant ($p = 0.79625$). A similarly statistically insignificant difference is found in the comparison of boars fed knotweed and gilts fed knotweed ($p = 0.09329$). Likewise, a statistically significant difference ($p = 0.00865$) between knotweed fed boars and gilts that were not fed on knotweed was found for serum albumin (Figure 6), which proves that knotweed deepens the difference between sexes in terms of this parameter. There was also a significant increase in concentrations between the first and second sampling in both

experimental and control groups and between the sexes, but in the gilts, the increase was higher, such that a statistically significant difference occurred in this group.

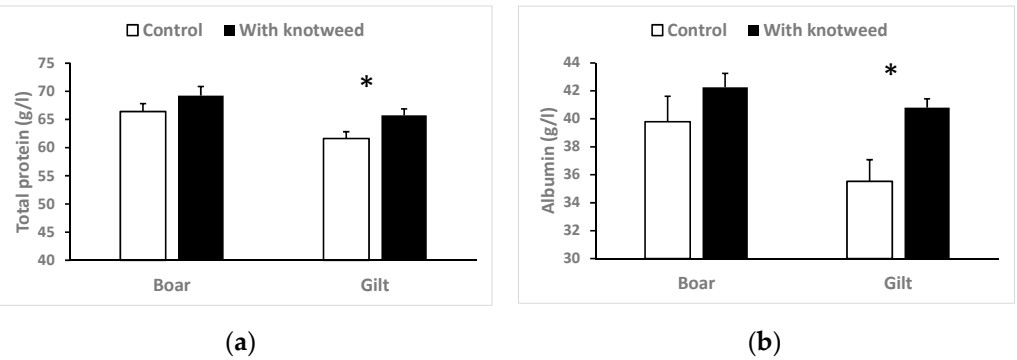

(**a**)                                    (**b**)

**Figure 6.** Total serum protein (**a**) and serum albumin (**b**) in the experimental group of pigs and the control group of pigs according to sex at the end of the experiment. Means + SEM. Difference significant on $p < 0.05$ marked by an asterisk.

### 3.4.2. Serum Cholesterol

There was a statistically significant difference in serum cholesterol (Figure 7); significantly lower values were recorded in the experimental group (fed on knotweed) compared to the control group.

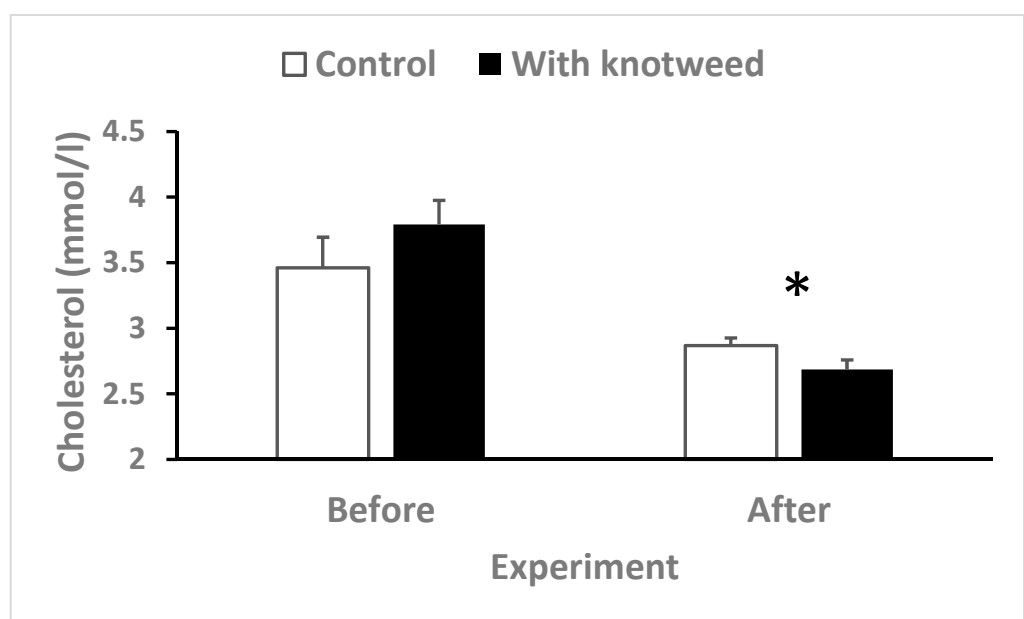

**Figure 7.** Serum cholesterol in the experimental and the control group at the beginning and the end of the experiment. Means + SEM. Difference significant on $p < 0.05$ marked by an asterisk.

### 3.4.3. Serum Creatinine

The experiment demonstrated a statistically significant change in the serum creatinine concentration; higher concentrations were recorded in the experimental group of pigs. Differences were obvious in both sexes, but a stronger reaction was recorded in gilts (Figure 8).

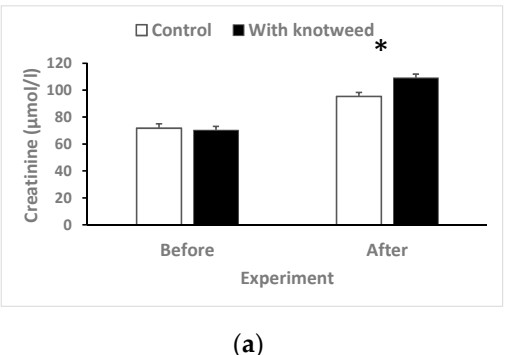 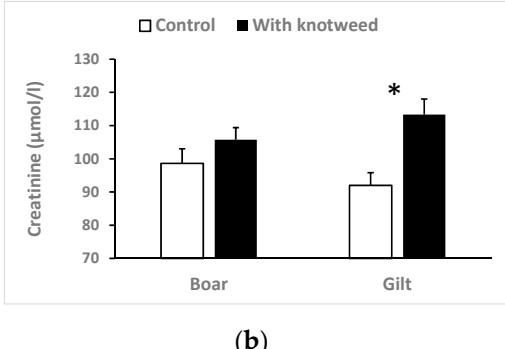

|     |     |
| :-: | :-: |
| (**a**) | (**b**) |

**Figure 8.** Serum creatinine in all animals at the beginning and the end of the experiment (**a**) and according to sex in the experimental group of pigs and the control group of pigs at the end of the experiment (**b**). Means + SEM. Difference significant on $p < 0.05$ marked by an asterisk.

### 3.4.4. Serum Glucose

From the results in Figure 9 it is obvious that the experimental pigs had lower concentrations of serum glucose, while, at the same time, even the differences between sexes were not statistically significant. Boars in the control group showed a significant increase in glucose, while boars fed on knotweed had serum glucose levels that were equivalent to those of gilts. Thus, it is possible to assume that the knotweed additive had a dampening effect on the increase in serum glucose concentration.

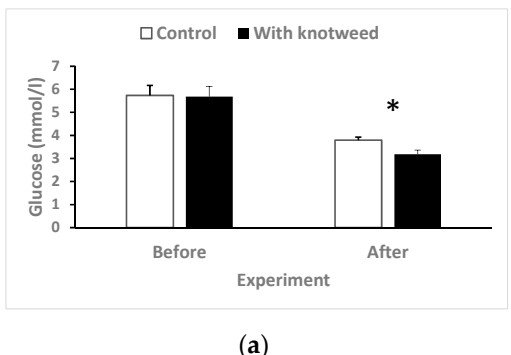 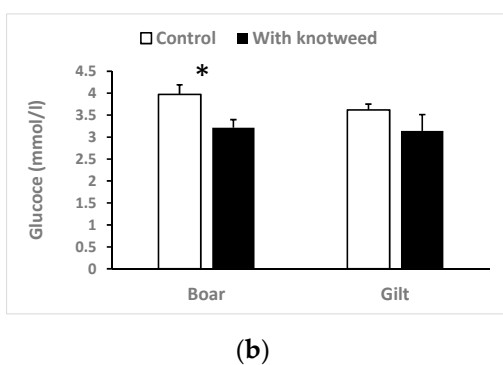

|     |     |
| :-: | :-: |
| (**a**) | (**b**) |

**Figure 9.** Serum glucose in all animals at the beginning and the end of the experiment (**a**) and according to the sex (**b**) in the tested group of pigs and the control group of pigs at the end of the experiment. Means + SEM. Difference significant on $p < 0.05$ marked by an asterisk.

### 3.4.5. Total Serum Bilirubin

After birth, the value of total serum bilirubin is generally higher because residues of fetal blood, which contains a higher number of immature cells, are being removed from the blood. After the subsequent decrease in the total serum bilirubin concentration, it is possible to see that the experimental group had a statistically significantly higher value of total serum bilirubin (Figure 10). More total serum bilirubin was found in boars, where the difference between the experimental and the control group was statistically significant compared to gilts (Figure 10).

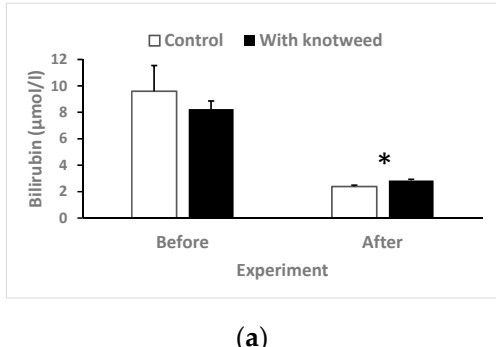 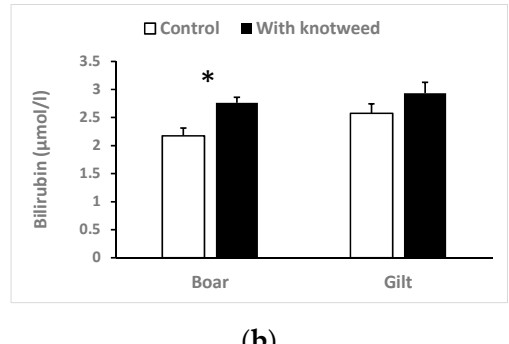

(**a**)         (**b**)

**Figure 10.** Total serum bilirubin in all animals at the beginning and the end of the experiment (**a**) and according to sex (**b**) in the experimental group of pigs (fed on knotweed) and the control group of pigs at the end of the experiment. Means + SEM. Difference significant on $p < 0.05$ marked by an asterisk.

### 3.4.6. Mean Cell Hemoglobin (MCH) and Mean Corpuscular Hemoglobin Concentration (MCHC) Indexes

From the results in Figure 11, it is obvious that the experimental group of pigs had significantly statistically higher levels of both MHC and MCHC compared to the control group.

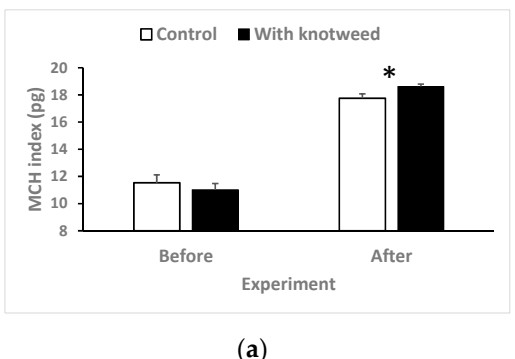 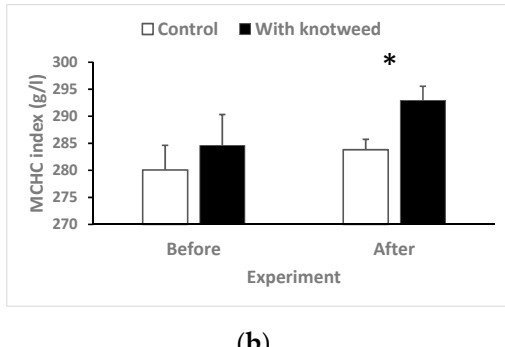

(**a**)         (**b**)

**Figure 11.** The MCH index (**a**) and the MCHC index (**b**) in all animals at the beginning and the end of the experiment in the experimental group of pigs fed on fodder with knotweed and the control group. Means + SEM. Difference significant on $p < 0.05$ marked by an asterisk.

### 3.4.7. Thrombocytes and Plateletcrit (PCT)

The experiment has proven a presumable effect of knotweed on the reduction of thrombocyte proliferation in the bone marrow (Figure 12); the control group had a statistically significantly lower number of thrombocytes. The PCT value (Figure 12) was in direct correlation to the number of thrombocytes; its value was also statistically significantly lower in the experimental group of pigs compared to the control group.

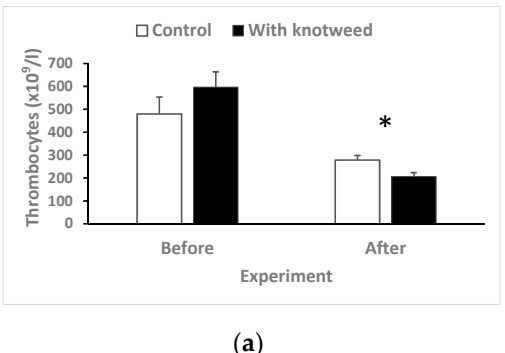 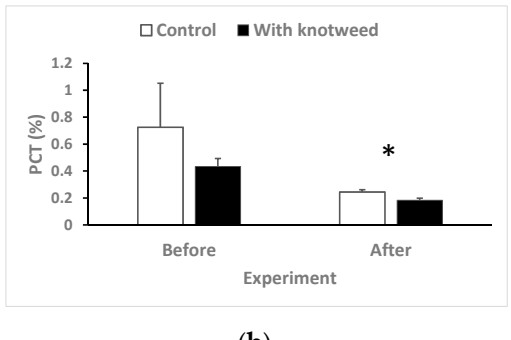

(**a**)　　　　　　　　　　　　　　　　(**b**)

**Figure 12.** Thrombocytes (**a**) and PCT (**b**) in all animals at the beginning and the end of the experiment in the experimental group of pigs fed on fodder with knotweed and the control group. Means + SEM. Difference significant on $p < 0.05$ marked by an asterisk.

### 3.4.8. Lymphocytes

Animals fed on knotweed showed an increase in the total number of lymphocytes in their blood (Figure 13). The control group recorded absolutely no progress in the number of lymphocytes. In the animals fed on knotweed, the number of white blood cells increased but the share of lymphocytes lowered. Nevertheless, the absolute values showed an increase in lymphocytes. Thus, the results represent a simple change of the percentage ratio between individual populations of white blood cells.

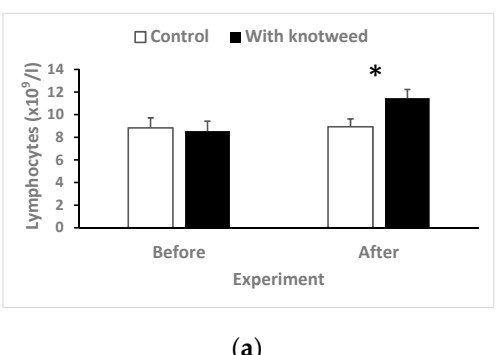 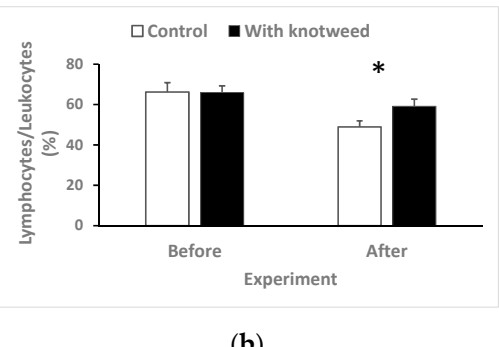

(**a**)　　　　　　　　　　　　　　　　(**b**)

**Figure 13.** The total number of lymphocytes (**a**) and the ratio of lymphocytes to leukocytes (**b**) in all animals at the beginning and the end of the experiment in the experimental group of pigs fed on fodder with knotweed and the control group at the end of the experiment. Means + SEM. Difference significant on $p < 0.05$ marked by an asterisk.

## 4. Discussion

### 4.1. Knotweed as a Fodder Additive

According to the results, it is obvious that the addition of knotweed into fodder improved the usability of fat in the EW (ČOS) fodder, dry matter in the A1 fodder and fibre in the A1 and KPB fodders. Similar to humans, pigs can only decompose fibre using intestinal bacteria. There is plenty of information about introducing fibre into pigs' diets. Jha and Leterme [60], using the in vitro fermentation method, stated that fibre may be important in the energy metabolism of a pig. Through its metabolites, fibre can directly stimulate physiological functions via local endocrine responses [61]. Residues from the production of such crops as rape [62] or corn [63] are also used as sources of fibre. Lindberg [64] stated that even though fibre is generally thought to impair the usability of nutrients in pigs, it is crucial for maintaining normal physiological functions of the digestive tract. Its properties are based on its source of origin. The scope of usability is based on the breed of pigs and their age. Oligo- and polysaccharides are being tested as prebiotics to improve pigs' immunity by affecting their intestinal microflora. Fibre

intake influences the expression of heat shock proteins in the intestinal epithelium, which influences the immune system of pigs.

### 4.2. Weight Changes

The Prestice Black-Pied breed is a combined meat–lard type with a thick layer of back fat [65]. The breed is characterised by a higher ratio of back fat and intramuscular fat, a lower share of muscle in the carcass and the top quality of meat [3]. Adding knotweed to the fodder increased the portion of back fat in gilts; it also increased the portion of muscle, but not as significantly as back fat and, on the contrary, it decreased the proportion of intramuscular fat. Currently, breeding is limited in the Czech Republic for pigs that have a lower share of lean meat [3], and the knotweed additive within our experiment lowered the Black-Pied's share of lean meat even further. According to Václavková [65], the breed also contains more fat compared to other genotypes of modern pigs. Thus, it seems that knotweed has a positive influence on growth, but only in gilts, and its influence is most likely related to the production of oestrogen. For instance, Szkudelska and Szkudelski [46] mention the ability of resveratrol to interact with oestrogen receptors.

The average daily weight growth in boars reached approx. 550 g, regardless of the fodder type. In the control group of gilts, the daily growth value of 400 g increased significantly to 500 g after using the fodder with knotweed. In our experimental breeding, which lasted approximately 175–184 days until the last weighing, we measured quite lower values than those given by Mátlová [66] for the same breed—525 g in gilts and 590 g in boars—or by Jedlička [67]—whose values of 562 and 584 g showed less difference between sexes. An increase in daily growth after the application of resveratrol was recorded by Zhang et al. [68] in intrauterine growth-retarded neonatal male piglets [Duroc × (Landrace × Yorkshire)].

Similar positive weight changes in gilts recorded by us are, on the contrary, in conflict with the results of some studies that state that resveratrol caused a reduction of body weight in obese mice and rats [49,69,70]. Similar to our results in pigs, no weight change upon the administration of knotweed to mice and rats was recorded by Baur et al. [71] and Rocha et al. [72].

### 4.3. Lipid Metabolism

In carcass of pigs weighing 86 kg, Alfonso et al. [73] measured the height of back fat (26 mm), with findings comparable to our results. Recently, many studies have shown that resveratrol has a body fat-lowering effect in rodents, but also some controversial studies reviewed by Zhang et al. [52] reported no body fat-lowering effect in humans and mice. Similarly, Meydani and Hassan [74] reviewed several animal studies showing that resveratrol treatments significantly reduced the size of fat depots in rodents. In a pig model, Zhang et al. [51], in contrast with our results, reported a decrease of back fat depth in barrows due to resveratrol. These different results may be caused by a whole range of factors. One of the reasons, in addition to the different breeds of pigs that were examined, may be that the studies used different doses of resveratrol, various feeding times and different methods of administering resveratrol. In our case, the fodder was supplemented with knotweed rhizomes, which also contain other secondary metabolites, while the cited studies used pure extracts of resveratrol. Most studies were also focused only on one sex or did not distinguish the sex during the evaluation of results, which can be misleading, as can be seen from the different responses of gilts and boars in our results.

In accordance with our results, Zhang et al. [51] reported a decrease of intramuscular fat content in barrows due to resveratrol that resulted in an improvement of pork meat quality. Meat quality is a complex trait influenced by many factors, including genetics, nutrition, feeding environment, animal handling and the interactions between these factors [75]. The content of intramuscular fat depends on the breed. Čechová et al. [76] found significantly lower portions of intramuscular fat, in the meat of the Czech Large White (0.62%), Czech Meaty Pig (0.98%) and Duroc (1.67%) breeds than we found in the Prestice Black-Pied

breed, even after the reduction caused by the knotweed. On the contrary, Alfonso et al. [73] recorded a 1% higher proportion of intramuscular fat in the carcasss of pigs weighing 86 kg. According to Yu et al. [77], the optimal intramuscular fat of pigs ranges between 2 to 3%, which is consistent with our results and the results of Kasprzyk et al. [78].

Nevrkla et al. [79] found slightly lower SFA content in both intramuscular fat ($36.76 \pm 2.95$ g $100$ g$^{-1}$) and back fat ($39.93 \pm 1.68$ g $100$ g$^{-1}$) in the Prestice Black-Pied breed. However, the proportion of PUFA was slightly higher ($12.97 \pm 3.37$, $13.52 \pm 2.83$ g $100$ g$^{-1}$, respectively). Similar values of the proportion of PUFA were also recorded by Serra et al. [80]. Ma and Sun [81] recorded the proportion of MUFA as $44.73 \pm 0.40$ g $100$ g$^{-1}$, which is comparable to our results, and found a slightly higher portion of PUFA ($13.97 \pm 1.63$ g $100$ g$^{-1}$) in the intramuscular fat. Similar values were also published by He et al. [82]. Čítek et al. [83] stated similar conclusions in terms of the differences between the representation of fatty acids in intramuscular and back fat. Polyunsaturated fatty acids are divided into n-3 and n-6. The n-3 FA is more beneficial to humans; they are thought to have a preventive effect against cardiovascular diseases and colon cancer and are important for the development of children's brains and vision [84,85]. Even though the MUFA/SFA, PUFA/SFA and n-6/n-3 PUFA ratios worsened, better values were still achieved than in the studies of Robina et al. [86] and Madzimure et al. [87]. Similar to our study, favourable ratios of fatty acids were also published by Nevrkla et al. [79].

The cholesterol content in the blood is closely related to lipid metabolism. Cholesterol is a fatty substance that is either received from food in the gut or produced in the liver when needed. It is involved in lipid transport in the blood between different tissues in the body [88]. Our study found a statistically significant reduction in the cholesterol content of the blood after the knotweed application in both sexes; a similar influence in pigs' models was reported by Elmadhun et al. [89]. Although Azorín-Ortuño et al. [90] did not confirm that consumption of resveratrol modified the level of cholesterol in mini-pigs, they found differences in the expression of some genes involved in lipid metabolism that were upregulated by the high-fat diet and downregulated by resveratrol. In accordance with our results, Azorín-Ortuño et al. [90] found that resveratrol and resveratrol-containing grape extract prevented the induction of fatty-acid binding proteins in peripheral blood mononuclear cells only in female mini pigs.

Many mechanisms have been described to explain the effect of resveratrol on lipid metabolism. Resveratrol may change the expression patterns of lipid metabolism-related genes and/or the activity of enzymes in muscle and adipose tissue [52]. Resveratrol reduced basal and insulin-induced glucose conversion to total lipids in white adipose tissue [46]. Resveratrol improved mitochondrial DNA content, ATP production and fatty acid oxidation in the liver of intrauterine growth-retarded mini piglets [68]. Resveratrol can increase the phosphorylation and activation of AMPK, the master regulator of energy metabolism, which upregulates fatty acid oxidation and increases the uptake of glucose through upregulation [74]. Cell culture studies have also shown that the addition of resveratrol can inhibit fatty acid and triglyceride synthesis in hepatocytes, can enhance lipolytic activity in adipocytes and can inhibit adipogenesis [74]. Serum leptin content was reduced by resveratrol [52], which is in line with the reduction of body fat mass because blood leptin level is positively associated with body fat mass [91]. Resveratrol can beneficially improve blood lipid profiles [52]. Resveratrol significantly inhibited pig preadipocyte proliferation and differentiation [92]. Supplemental resveratrol positively influenced glucose metabolism pathways in the liver and skeletal muscle and led to improved glucose control in a swine model of metabolic syndrome [93].

### 4.4. Biochemistry and Haematology

Total serum protein is the sum of all protein fractions in the serum, i.e., albumin and globulins. Its concentration is influenced by several factors such as the amount and representation of amino acids in a fodder's proteins, the current needs of the organism, protein loss and relative changes, i.e., loss or intake of fluids [94,95]. The ratios of serum

albumin and serum globulins may be different, and if, for example, a reduction of serum albumin concentrations occurs and the concentration of serum globulins is competitively increased, this may not affect the total serum protein concentration [96]. The effect of knotweed on the total serum protein content was proven by our experiment, and moreover, it was proven that knotweed positively influenced gilts to increase the TSP concentration to the same level as boars.

Serum albumin is a protein in the serum and plasma synthesized by the liver. Its main function is to maintain oncotic pressure, transfer various substances and act as a protein store. Serum albumin is just one of the hundreds of proteins that contribute to the concentration of total serum protein. However, its share is the largest by far of all proteins [96]. Since the second component of the total serum protein—serum globulin—did not change statistically between the experimental and control groups during the experiment, it is obvious that serum albumin is the protein part that led to the statistically significant change in the total serum protein concentration. Increased serum albumin production is initiated by the liver as needed upon the decrement of oncotic pressure, eventually due to thyroid hormones, corticosteroids, growth hormone or insulin [95]. It is not known by what mechanism the increase in serum albumin concentration in the tested pigs occurred; however, similar results were also recorded by a study evaluating the safe use of pure resveratrol in laboratory animals [97]. In our experiment, the increase of serum albumin may be related to the fact that free fatty acids are connected to it and transported to target tissues.

Serum creatinine is an end product of creatine metabolism in muscles that is further transported to the kidneys through the blood, from which it is excreted in the urine without threshold via glomeruli and partly also via tubules. At physiological plasma concentrations, urinary excretion is stable, i.e., not dependent on feeding [94]. There is still an ongoing debate as to how serum creatine concentration can be increased within physiological limits. An initial opinion that claimed that the amount of produced creatinine corresponds proportionally to the muscle mass of the individual was rejected following the publication of human studies that showed that serum creatinine concentration was influenced by age and sex [98]. However, recent studies lean back towards the initial opinion that creatinine concentrations in plasma/serum are related to the volume of muscle mass. A new method has even been developed to estimate the volume of human muscle mass based on the serum concentration of creatinine and cystatin C [99]. Our results also correlate with this opinion; the statistically significant increase in the creatinine concentration in the gilts' serum can be related to the increased muscle growth in the tested animals in contrast with the control group.

Serum glucose represents a quick energy source for cells. Its concentration may vary considerably, depending not only on the time that has elapsed since feeding but the experience of stress due to, for instance, manipulation or blood sampling. However, serum glucose concentrations also change rapidly depending on sample handling conditions; it decreases in whole blood after 1 h by about 0.5 mmol/L [94]. Thus, results should be evaluated very cautiously. From the results, it is obvious that pigs fed on knotweed had a lower concentration of serum glucose, and there was no statistically significant difference between the sexes. This result can be considered consistent with the results of other studies that have proven the depressive influence of substances contained in knotweed—namely resveratrol—on the concentration of serum glucose [89,93,100]. A study has even proven that piceid, a synonym for polydatin, reduced serum glucose concentrations in mice with experimentally induced diabetes mellitus [101]. In our study, boars in the control group showed a significant increase in serum glucose, while boars fed on knotweed had a level of serum glucose that was equivalent to that of the gilts. Therefore, it can be presumed that the knotweed additive had a damping effect on the increase of the serum glucose concentration.

Serum bilirubin is an intermediate product of the haemoglobin degradation released during the breakdown of red blood cells. This process occurs naturally and constantly due to age, but haemoglobin may also release more rapidly during haemolytic processes or

from haemorrhages. In addition to haemoglobin, substances containing porphyrin groups are also degraded to bilirubin, e.g., myoglobin, cytochrome 450, peroxidase, catalase, etc. Hepatic hyperbilirubinemia in acute or chronic processes are related to the destruction of or defects in hepatocytes, or cholestatic hyperbilirubinemia is possible due to the rupture of a gallbladder or the main bile duct [96]. The effect of resveratrol on the concentration of total bilirubin in plasma was tested within several human studies [102] in which the experimental group showed no change in the concentration of total plasma bilirubin but a statistically significant difference occurred in the control group, where the concentration of bilirubin decreased. The effect of resveratrol on increasing plasma bilirubin concentrations has been proven at an extremely high dose—3000 mg/kg of live weight in laboratory rats; no significant difference occurred at lower doses [103]. That knotweed increases serum bilirubin can be considered proven, but the course and causes of the reaction remain unknown.

Thrombocytes are fragments of megakaryocytes without a core and with a number of organelles in the cytosol. They play a crucial role in blood clotting, and they are responsible for the initial phase of stopping the blood flow to the microvascular plexus at the site of injury or damage [104]. They are created, like most other blood cells, in the bone marrow [105]. Their number is influenced by the rate of creation, consumption and loss [104]. In previous studies, it was proven that the administration of resveratrol increased the digestion abilities of thrombocytes and caused a significant decrease in the number of thrombocytes in the peripheral blood [106]. This decrease was also observed in our study. In parallel, we also observed the reduction of the thrombocrit value, i.e., PCT. The mechanism of this phenomenon is unknown. One possibility is the depressive influence of some substances contained in knotweed, e.g., resveratrol. Resveratrol has been proven to inhibit the growth and the induction of apoptosis in both normal and leukemic cells in the bone marrow [32]. The specific apoptotic effect of resveratrol against thrombocytes was also affirmed by [107].

Blood lymphocytes comprise different subpopulations with various immune functions, but for their identification and quantification, it is necessary to execute specific examinations, e.g., according to the cluster of differentiation (CD) system on the cells' surface. CD molecules are markers of the cell surface, which could be used for cell-type identification. The functions of CD molecules are various. The effect of resveratrol on the number of lymphocytes has been studied in vitro on cell cultures. While a high concentration of resveratrol caused the inhibitory proliferation of lymphocytes, low resveratrol concentrations of 0.75–6.00 µmol/L increased production [108,109]. The actual effects of resveratrol with respect to in vivo knotweed can only be estimated; however, no certainty can be obtained without a properly conducted study.

## 5. Conclusions

The addition of knotweed to the fodder of Prestice Black-Pied pigs stimulated a whole range of physiological changes. It positively stimulated weight growth and increased the back fat and proportion of muscle, but this was statistically significant only in gilts. As a result, gilts fed on knotweed had almost the same weight as boars by the end of the experiment. However, in terms of fatty acids, the knotweed additive seems to be unsatisfactory because it increased the proportion of SFA and decreased the proportion of PUFA, especially n-3 PUFA, which also influenced the n-6/n-3 ratio, which should optimally be 4:1. According to available data, our research is the first to study the effects of knotweed fodder additives on pigs' development, and it will be necessary to conduct other more detailed research focused on the confirmation of the observed changes and especially their explanation.

## 6. Patents

The presented study is a part of a more extensive project resulting in the registration of the Utility Model "Compound feed containing knotweed for game, domestic and farm animals".

**Author Contributions:** Conceptualization, P.M., M.B., M.K. and J.N.; methodology, M.R., J.T., E.V., M.K., P.M., J.N.; formal analysis, T.F.; investigation, J.T., E.V., Š.V., S.K., J.V., Z.N.; data curation, T.F., J.V., Z.N.; writing—original draft preparation, P.M., Š.V., L.M., R.F.; writing—review and editing, all authors; visualization, T.F. All authors have read and agreed to the published version of the manuscript.

**Funding:** This research was funded by Technology Agency of the Czech Republic, grant number TH02010325" within project "Innovation of feed supplements for fitness improvement of domestic and wild animals".

**Institutional Review Board Statement:** Not applicable.

**Informed Consent Statement:** Not applicable.

**Data Availability Statement:** Data are available by corresponding author.

**Conflicts of Interest:** The authors declare no conflict of interest.

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
