# Peer review of "Effect of Knotweed in Diet on Physiological Changes in Pig"

_agriculture, doi:10.3390/agriculture11020169_

Round 1

Reviewer 1 Report

I have only read the first part of the paper and, already in the materials and methods, the authors should better clarify how they conducted the trial.

The feed intake of the animals is not adequately specified and therefore it is not clear how many metabolites (especially resveratrol) the pigs ingested daily.

If I did the calculations correctly and if I understood correctly, according to what the authors stated, for 1 kg of food supplemented with knotweed, 0.162 mg of resveratrol ingestion correspond.

If so, this dose is very far, in my opinion, from the one potentially capable of causing effects on the metabolism of animals.

For example, it is much lower than that used in other research cited by the authors

(46. Cheng, Z.; Junqiu, L.; Bing, Y.; Ping, Z.; Zhiqing, H.; Xiangbing, M.; Jun, H.; Jie, Y.; Jiali, Ch.; Daiwen, Ch. Dietary resveratrol 784 supplementation improves meat quality of finishing pigs through changing muscle fibre characteristics and antioxidative 785 status. Meat Science 2014, 102, 15–21. https://doi.org/10.1016/j.meatsci.2014.11.014. 786

47. Cheng, Z.; Junqiu, L.; Bing, Y.; Jiali, Ch.; Daiwen, CH. Effects of resveratrol on lipid metabolism in muscle and adipose tissues: 787 A reevaluation in a pig model. Journal of functional foods 2015, 14, 590–595. https://doi.org/10.1016/j.jff.2015.02.039 )

where administrations of 300, 600 mg / d of resveratrol are used.

Much lower even than those reported by Azorin-Ortuno et al (2012)

(87 Azorín-Ortuño, M.; Yáñez-Gascón, M.J.; González-Sarrías, A.; Larrosa, M.; Vallejo, F.; Pallarés, F.J.; Lucas, R.; Morales, J.C.; 879 Tomás-Barberán, J.A.; García-Conesa, M-T.; Espín, J.C. Effects of long-term consumption of low doses of resveratrol on di-880 et-induced mild hypercholesterolemia in pigs: a transcriptomic approach to disease prevention. Journal of Nutritional Biochem-881 istry 2012, 23, 829–837. https://doi.org/10.1016/j.jnutbio.2011.04.007 )

where the LOW level of resveratrol used was equal to 8 mg / d.

So, for this reason alone, I suspend my judgment on the paper, waiting for the authors to clarify the doses used in the trial.

I would also suggest:

- better specify the statistical analysis that was carried out

- report the data indicated in the many graphs presented in tables that would be more readable. Estimated means should be presented with standard error and not standard deviation, with the significance of any differences found.

- in the references, paper no. 46 is the same as paper 71, while paper no. 47 is the same as No. 69.

Author Response

Q: English language and style

(x) Extensive editing of English language and style required

Response: The English was revised by special company using native speaker (certificate attached)

Comments and Suggestions for Authors

Q: I have only read the first part of the paper and, already in the materials and methods, the authors should better clarify how they conducted the trial.

Response: Some parts of MaM section were clarified.

Q: The feed intake of the animals is not adequately specified and therefore it is not clear how many metabolites (especially resveratrol) the pigs ingested daily.

If I did the calculations correctly and if I understood correctly, according to what the authors stated, for 1 kg of food supplemented with knotweed, 0.162 mg of resveratrol ingestion correspond.

If so, this dose is very far, in my opinion, from the one potentially capable of causing effects on the metabolism of animals.

For example, it is much lower than that used in other research cited by the authors

(46. Cheng, Z.; Junqiu, L.; Bing, Y.; Ping, Z.; Zhiqing, H.; Xiangbing, M.; Jun, H.; Jie, Y.; Jiali, Ch.; Daiwen, Ch. Dietary resveratrol 784 supplementation improves meat quality of finishing pigs through changing muscle fibre characteristics and antioxidative 785 status. Meat Science 2014, 102, 15–21. https://doi.org/10.1016/j.meatsci.2014.11.014. 786

  1. Cheng, Z.; Junqiu, L.; Bing, Y.; Jiali, Ch.; Daiwen, CH. Effects of resveratrol on lipid metabolism in muscle and adipose tissues: 787 A reevaluation in a pig model. Journal of functional foods 2015, 14, 590–595. https://doi.org/10.1016/j.jff.2015.02.039 )

where administrations of 300, 600 mg / d of resveratrol are used.

Much lower even than those reported by Azorin-Ortuno et al (2012)

(87 Azorín-Ortuño, M.; Yáñez-Gascón, M.J.; González-Sarrías, A.; Larrosa, M.; Vallejo, F.; Pallarés, F.J.; Lucas, R.; Morales, J.C.; 879 Tomás-Barberán, J.A.; García-Conesa, M-T.; Espín, J.C. Effects of long-term consumption of low doses of resveratrol on di-880 et-induced mild hypercholesterolemia in pigs: a transcriptomic approach to disease prevention. Journal of Nutritional Biochem-881 istry 2012, 23, 829–837. https://doi.org/10.1016/j.jnutbio.2011.04.007 )

where the LOW level of resveratrol used was equal to 8 mg / d.

So, for this reason alone, I suspend my judgment on the paper, waiting for the authors to clarify the doses used in the trial.

Response: Sorry for mistake, during translation the value of resveratrol content was deleted, we added this value again and we also count average daily dose of resveratrol per animal in experimental group.

I would also suggest:

Q: - better specify the statistical analysis that was carried out

Response: The statistical analyses were described in more details.

Q: - report the data indicated in the many graphs presented in tables that would be more readable. Estimated means should be presented with standard error and not standard deviation, with the significance of any differences found.

Response: Data of four figures are presented in Table 1 now. The standard errors of mean were already mentioned in figures, now standard deviations were replaced by SEM in the tables and in the text, Also significance of any differences is mentioned in manuscript.

Q: - in the references, paper no. 46 is the same as paper 71, while paper no. 47 is the same as No. 69.

Response: Thanks for reporting this mistake, it has been fixed.

Reviewer 2 Report

In this study, authors examined the effects of the addition of Bohemian knotweed to the diet on growth and blood haematology and biochemistry in pigs. The results showed the significant alterations in growth but only in gilts. Moreover, there was a significant changes in several parameters such as fatty acid composition and blood biochemicals. The manuscript seems well-written, however, there are still some issues described below.

  1. Abstract

Authors should describe the concentration of added knotweed in the diet in the abstract.

  1. Feed (p.3 line 103-129)

Authors should describe the information about Dibaq (e.g. country and prevalence of the feeds made by this company).

  1. Abbreviations

Authors used many abbreviations without their definitions. Define the below terms.

  1. 3 line 107, GM
  2. 3 line 126, FC
  3. 9 Table1, MLLT
  4. 12 line 403, MCH and MCHC
  5. 12 line 413, PCT
  6. 15 line 534, RES
  7. 17 line 643, CD

  1. p. 4 line 163

K3EDTA -> EDTA3K

  1. Figures 1-17

Insert the index next to X-axis in addition to its unit in all figures (e.g. Dry matter (%) in Fig.1(a)).

  1. Figure legends

Describe the number of pigs in all figure legends. This should be written as “Means plus/minus SEM for (digit) pigs”.

  1. p. 6 line 243 and p.7 line 245

Add SEM to the values of intramuscular fat.

  1. p. 9 line 318

Clearly state that the these parameters were measured using blood samples. For example, use total blood protein (or serum or plasma according to the condition) instead of total protein. The all biochemical parameters using blood should be modified. These modifications should be also done in figure legends (Fig.9-14).

  1. p. 13 line 460, p. 14 line 486

carcase -> carcass?

Author Response

Q: English language and style

 (x) English language and style are fine/minor spell check required

Response: The English was revised by special company using native speaker (certificate attached)

Comments and Suggestions for Authors

In this study, authors examined the effects of the addition of Bohemian knotweed to the diet on growth and blood haematology and biochemistry in pigs. The results showed the significant alterations in growth but only in gilts. Moreover, there was a significant changes in several parameters such as fatty acid composition and blood biochemicals. The manuscript seems well-written, however, there are still some issues described below.

  1. Abstract

Q: Authors should describe the concentration of added knotweed in the diet in the abstract.

Response: Requested information was added into abstract.

  1. Feed (p.3 line 103-129)

Q: Authors should describe the information about Dibaq (e.g. country and prevalence of the feeds made by this company).

Response: Requested information was added.

  1. Abbreviations

Q: Authors used many abbreviations without their definitions. Define the below terms.

  1. 3 line 107, GM
  2. 3 line 126, FC
  3. 9 Table1, MLLT
  4. 12 line 403, MCH and MCHC
  5. 12 line 413, PCT
  6. 15 line 534, RES
  7. 17 line 643, CD

Response: The abbreviations were defined directly in the text in the first place where they are mentioned

  1. p. 4 line 163

Q: K3EDTA -> EDTA3K

Response: Fixed.

  1. Figures 1-17

Q: Insert the index next to X-axis in addition to its unit in all figures (e.g. Dry matter (%) in Fig.1(a)).

Response: Index was added in all graphs as recommended.

  1. Figure legends

Q: Describe the number of pigs in all figure legends. This should be written as “Means plus/minus SEM for (digit) pigs”.

Response: Number of pigs in control and experimental groups are mentioned in Methodology. We believe that this does not need to be repeated for each Figure.

  1. p. 6 line 243 and p.7 line 245

Q: Add SEM to the values of intramuscular fat.

Response: SEM values were added.

  1. p. 9 line 318

Q: Clearly state that the these parameters were measured using blood samples. For example, use total blood protein (or serum or plasma according to the condition) instead of total protein. The all biochemical parameters using blood should be modified. These modifications should be also done in figure legends (Fig.9-14).

Response: Clarification was done through entire manuscript.

  1. p. 13 line 460, p. 14 line 486

Q: carcase -> carcass?

Response: The term was unified as carcass through entire manuscript. 

Reviewer 3 Report

Dear Authors,

The manuscript (agriculture-1107792) presented for review is very interesting and I recommend the article for publication in Agriculture Journal after major revision in sections Abstract, Introduction, and Results. 

I believe it addresses an important area of research in an international context.

Reviewer

Author Response

Q: Title: Effect of Knotweed in Pig Diet, rather should be changed, for example to: Effect of Knotweed in Diet on physiological changes in pig, or similar title.

Response: Thanks for this suggestion, it was accepted,

Q: Abstract: In my opinion Abstract sounds a little bit like Introduction, but I reckon that should has the following structure: contain the aim of manuscript, information about contain and the conclusions. The abstract does not present the obtained results, it is too general.

Response: We wrote abstract exactly according to the template provided by editorial office and limit of 200 words doesn´t allow us to mention results in more detail.

Q: Introduction: Genetic advances in livestock breeding in recent years have significantly increased their production potential. Currently, animals are characterized of high productivity, and at the same time are very demanding in terms of nutrition and housing conditions. This applies to pig breeds with high genetic potential. In order to exploit their genetic potential, pigs require a specific nutrition, which covers the high nutrient requirements. That can only be provided by a feed containing the right amount and quality of protein, energy, minerals, vitamins and feed additives.

Response: Prestice Black-Pied pig is old fashioned traditional Czech pig breed with lower productivity.

Q: In my opinion, in the Introduction section, Authors should write a little about this specific breed, which is Prestice Black-Pied pigs, and about the problems with its feeding, if they are looking for new ideas for their diet and dietary additives.

Response: Thanks for recommendation, we described Prestice Black-Pied breed shortly in the Introduction.

Q: What effect did the authors of the studies want to achieve? I reckon, that not only whether this effect would be positive or negative? Why did they think this supplement would be beneficial? What were the hypotheses?

Response: The aim of study and expected results were specified in last paragraph of Introduction.

Q: Was knotweed also considered in terms of environmental risk, as it effectively competes with native plant species, hindering their growth and regeneration? It is an invasive species, and its cultivation is prohibited in some countries, as the authors wrote about in section Introduction. If it proves beneficial as a dietary additives in pigs feeding, where will it be obtained from?

Response: It is a very good question in case of potential commercial application of our results. There is possible to obtain an exception for knotweed cultivation within arable land from Nature conservation authorities in the Czech Republic. But we think, this issue goes beyond the focus of the article.

Q: The authors of the manuscript refer in sections Introduction and Discussion to resveratrol, but in many studies it was used as a pure substance, not from a raw material, in which it is in various types of relationships. The literature review does not show whether the remaining compounds of knotweed were added to animal feed by other researchers. Authors, Please present the research gap and why that this research is so important.

Response: The literature review already contains some papers focused on effect of piceid or emodin, but we added more as requested.

Material and Methods

Q: 2.3.6 Statistical data evaluation – Details of the statistical tests used and the level of significance are lacking.

Response: The statistical analyses were described in more details and level of signifikance was added to the M&M section.

Results

Q: All Figures do not have a description of the X and Y axes. It seems that some Figures could be combined, for example, 5 and 7, 6 and 8, taking into account a different bars color for IMF and lard.

Now, It difficult to understand what present the Figure, because each is similar.

Response: Data of the figures mentioned are presented in Table 1. Description of Y axes was completed. We believe description of X axes is adequate.

Q: 3.4.1. Total protein/ 3.4.2 Cholesterol  – Neither in the description nor in the figures it is stated that contents of protein and cholesterol is connected with the blood of pigs, which may mislead the readers.

Response: It was specified as requested also by reviewer 2.

Q: References –  I am impressed with the Reference list of articles, but half of these references come from 2000-2010 years, and 11 of references come from before 2000. In the last 20 years the animal breeding and feeding was change. Moreover, Authors Please check the correctness of the citation of references by requirements of the Agriculture Journal. I have impression that in some of reference authors didn't write properly.

Response: References were checked.

Round 2

Reviewer 1 Report

The authors responded adequately to the observations made in the first review of the paper. The authors should, to further clarify the results obtained, insert the data relating to the feed intake of fodder, during the trial, of the experimental groups.

Author Response

We added Table 1 into the section "3.1. Fodder consumption and digestibility" where the requested fodder consumption is shown. The other Tables were re-numbered.

Reviewer 3 Report

Dear Authors,

The authors have changed many parts of the planned paper number agriculture-1107792 according to my suggestions.

I would like to thank the authors for considering my comments and applaud them for the major revisions to improve their manuscript.

The manuscript is interesting and valuable.

Reviewer

Author Response

The reviewer didn´t insert any comments or requests for revision.